# Exploring the Potential Mechanism of Action of Piperine against *Candida albicans* and Targeting Its Virulence Factors

**DOI:** 10.3390/biom13121729

**Published:** 2023-11-30

**Authors:** Claudia Patricia Bravo-Chaucanés, Luis Carlos Chitiva, Yerly Vargas-Casanova, Valentina Diaz-Santoyo, Andrea Ximena Hernández, Geison M. Costa, Claudia Marcela Parra-Giraldo

**Affiliations:** 1Unidad de Proteómica y Micosis Humanas, Grupo de Enfermedades Infecciosas, Departamento de Microbiología, Facultad de Ciencias, Pontificia Universidad Javeriana, Bogotá 110231, DC, Colombia; claub06@gmail.com (C.P.B.-C.); y.vargasc@javeriana.edu.co (Y.V.-C.); valentina_diaz@javeriana.edu.co (V.D.-S.); 2Grupo de Investigación Fitoquímica Universidad Javeriana (GIFUJ), Departamento de Química, Facultad de Ciencias, Pontificia Universidad Javeriana, Bogotá 110231, DC, Colombia; chitival@javeriana.edu.co (L.C.C.); hernandez_a@javeriana.edu.co (A.X.H.); modesticosta.g@javeriana.edu.co (G.M.C.); 3Departamento de Microbiología y Parasitología, Facultad de Farmacia, Universidad Complutense de Madrid, 28040 Madrid, Spain

**Keywords:** antifungal activity, *Candida albicans*, mechanism of action, piperine, virulence factors

## Abstract

Plant-derived compounds have proven to be a source of inspiration for new drugs. In this study, piperine isolated from the fruits of *Piper nigrum* showed anti-*Candida* activity. Furthermore, the mechanisms of action of piperine and its impact on virulence factors in *Candida albicans*, which have not been comprehensively understood, were also assessed. Initially, piperine suppressed the hyphal transition in both liquid and solid media, hindered biofilm formation, and resulted in observable cell distortions in scanning electron microscope (SEM) samples, for both fluconazole-sensitive and fluconazole-resistant *C. albicans* strains. Additionally, the morphogenetic switches triggered by piperine were found to rely on the activity of mutant *C. albicans* strains. Secondly, piperine treatment increased cell membrane permeability and disrupted mitochondrial membrane potential, as evidenced by propidium iodine and Rhodamine 123 staining, respectively. Moreover, it induced the accumulation of intracellular reactive oxygen species in *C. albicans*. Synergy was obtained between the piperine and the fluconazole against the fluconazole-sensitive strain. Interestingly, there were no hemolytic effects of piperine, and it resulted in reduced cytotoxicity on fibroblast cells at low concentrations. The results suggest that piperine could have a dual mode of action inhibiting virulence factors and modulating cellular processes, leading to cell death in *C. albicans*.

## 1. Introduction

*Candida* is a yeast that naturally exists in the human microbiota as a commensal organism [1]. However, under certain favorable conditions or predisposing circumstances, it can transform an opportunistic pathogen, giving rise to various types of infections, which can span from superficial to systemic and can manifest in various parts of the body. Notably, candidiasis can lead to substantial morbidity and mortality, particularly among individuals with compromised immune systems [2,3]. Currently, there are some broad-spectrum antifungal drugs accessible for treating infections caused by *Candida*. Nonetheless, the emergence of adverse side effects and the growing problem of antifungal resistance have raised concerns regarding the efficacy of existing antifungal treatments. Consequently, a pressing demand exists for novel and enhanced antifungal agents to effectively tackle these issues [4].

An innovative approach in the search for *Candida* inhibitors is to target the virulence factors of the pathogen. This strategy aims to disable *Candida’s* ability to infect without killing it, thus preventing the emergence of drug-resistant mutants [5]. In particular, the ability of *Candida albicans* to undergo a reversible morphological transition is crucial for establishing pathogenicity during an infection by evading macrophage phagocytic activity and enhancing epithelial cell invasion [6,7]. Additionally, it serves as a critical factor in both the development and preservation of biofilm structure on host tissue [8]. Biofilm formation by *C. albicans* is regarded as another significant virulence factor, as sessile cells within the biofilm display heightened resistance to antifungal agents [9]. As a matter of fact, a significant number of *C. albicans* infections are linked to the development of biofilms on both host and abiotic surfaces, including medical devices [10].

On the other hand, investigating the modes of action of antifungal agents is a fundamental strategy aimed at constraining the development of resistance to existing agents while also facilitating the development of safer and more efficacious compounds [11]. The acknowledgment of the significance of programmed cell death in fungi has prompted a reassessment of the mechanism of action of antifungal agents [12]. Apoptosis in fungi can be triggered by various factors and is accompanied by most of the typical features associated with multicellular apoptotic cell death, such as nuclear condensation, DNA fragmentation, the accumulation of reactive oxygen species (ROS), an increase in mitochondrial permeability, and cell cycle arrest [13].

In recent centuries, plants have been recognized as a valuable resource for the discovery of new drugs. The presence of secondary metabolites with diverse biological activities, including antimicrobial properties, make plants a promising source for drug development as they have demonstrated the ability to inhibit fungal cell wall, and sphingolipid and protein synthesis [14]. As our lab reported previously, the ethanolic extract of *Piper nigrum* showed a potential anti-virulence against *C. albicans* [15]. This extract contained piperine, an important alkaloid that has been found to have a wide range of pharmacological activities [16,17]. This secondary metabolite holds the potential for therapeutic applications. Exploring its biological activity against *C. albicans* would be of significant interest.

Based on some reported studies, piperine has been shown to have antifungal properties against *Candida* spp. [18] as well as antibacterial activity against human pathogens [19,20]. Currently, certain findings indicate that piperine may participate in the regulation of morphological transitions and biofilm development [21]. However, it is important to note that reference data in this area are often limited or contradictory. A recent study showed that piperine has synergistic activity with fluconazole (FLC) against the growth of an FLC-resistant strain of *C. albicans* [22]. Combinatorial drug therapy has garnered significant attention owing to its multifaceted benefits, including reduced toxicity, improved efficacy, and the absence of antibiotic resistance [23]. The capacity of piperine to destabilize membranes also positions it as a promising contender for addressing drug-resistant *C. albicans* infections and biofilms, where achieving a threshold concentration of FLC is challenging [22]. As a significant constituent of pepper, piperine contributes, at least in part, to these activities. Hence, further studies should prioritize investigating the specific effects of pure piperine to gain a deeper understanding of its potential.

This study aimed to investigate piperine against FLC-sensitive (SC5314) and FLC-resistant (CAAL256) strains of *C. albicans.* The primary focus was to evaluate piperine’s anti-*Candida* activity, particularly inhibiting some virulent features such as biofilm formation and morphological transitions resulting in observable changes in cell morphology. Additionally, the study aimed to elucidate the mechanisms behind piperine’s effects on *C. albicans*, including its influence on membrane integrity, the regulation of reactive oxygen species, mitochondrial function, and the potential synergistic effects of piperine and FLC against *C. albicans*. The ultimate goal was to explore piperine as a natural alternative for antifungal treatment.

## 2. Materials and Methods

### 2.1. Common Experimental Procedures

Thin-layer chromatography (TLC) was conducted using Merck alumina silica gel 60 F_254_ plates. The melting point was recorded using a Thermo-Scientific 00590Q Fisher-Johns apparatus. Vacuum liquid chromatography (VLC) was performed using silica gel with a particle size of 230–400. Purification was carried out on a CPC-250 equipment (Gilson^®^, Limburg, Germany). Nuclear magnetic resonance (NMR) measurements were conducted on a Bruker Avance III HD 600 MHz spectrometer (Bruker, Millerica, MA, USA). The 1D and 2D experiments were recorded at 600 MHz for ^1^H and 150 MHz for ^13^C using the solvent peaks as internal references. The spectra were recorded in deuterated chloroform 99.8 atom % deuterium (CDCl_3_). High-resolution mass spectrometry (HRMS) analysis was conducted using Shimadzu Nexera LCMS 9030 equipment. The ionization method operated with electrospray in positive mode (ESI^+^).

### 2.2. Phytochemical Extract

*Piper nigrum* L. was collected in Orito, in the department of Putumayo, Colombia (0°37′45′′ N, 76°51′55′′ W). The plant material was identified, and a voucher (HPUJ-30548) was deposited, as previously reported [15]. 500 g of dried and pulverized fruits of *P. nigrum* was extracted via percolation using 96% ethanol at a plant-to-solvent ratio of 1:10 (*w*:*v*). Four cycles of 24 h each were conducted, and the extracts obtained from each cycle were pooled and dried through a rotary evaporation process, resulting in the respective extract. 

### 2.3. Purification of Piperine by Centrifugal Partition Chromatography (CPC)

50 g of the obtained extract was previously fractionated by VLC with solvents in increasing polarity, resulting in the fractions n-hexane (2.5 g), dichloromethane (25 g), ethyl acetate (5 g), and ethanol–water (1 g). In a previous High-Performance Liquid Chromatography with Diode-Array Detection (UPLC-DAD) analysis, it was determined that the dichloromethane fraction contained a major compound identified as piperine [15], so this fraction was selected for purification on CPC. Initially, the solvent system was selected by dissolving a small amount of the dichloromethane fraction in test tubes containing variations of the hexane–ethyl acetate–methanol–water (HEMWat) system. Subsequently, the test tubes were shaken for 60 s and once rested and the sample was partitioned between phases, equal volumes of each phase was monitored by TLC. The selection of the solvent system was based on the visual distribution of the analyte between the upper phase (UP) and the lower phase (LP), where the system consisting of HEMWat was in a ratio of 6:5:6:5 *(v*/*v*/*v*/*v)* chosen for the CPC separation. A descending separation mode was employed, where the UP served as the stationary phase, while the LP acted as the mobile phase. Column filling with stationary phase was performed at a rotation speed of 500 rpm and a flow rate of 25 mL/min. Mobile phase was pumped at 1500 rpm and 5 mL/min. After reaching hydrodynamics equilibrium, the retention of the stationary phase was calculated as 70%. Then, a sample of 2 g from the dichloromethane fraction, diluted in 5 mL of UP and 5 mL of LP, was injected into a 10 mL loop to obtain 500 mg of compound **1** for biological test [24,25]. Fractions were collected and monitored by TLC, as well as pooled according to their chromatographic profile, resulting in the isolation of compound **1**. The identification of compound **1** was performed using ^1^H NMR, ^13^C NMR (DEPTQ), COSY, HSQC, and HMBC, and HRMS techniques; also, it was complemented with the compared data in the existing literature (Appendix A).

Piperine (**1**). Yellow crystalline solid, m.p. 127–129 °C. ^1^H NMR (600 MHz, Chloroform-*d*): δ (ppm) 7.36 (ddd, *J* = 14.7, 9.0, 1.2 Hz, 1H), 6.94–6.92 (m, 2H), 6.85–6.80 (m, 3H), 6.40 (d, *J* = 14.7 Hz, 1H), 5.94 (s, 1H), 3.60–3.42 (m, 4H), 1.67–1.59 (m, 4H), 1.56–1.48 (m, 2H). DEPTQ (150 MHz, Chloroform-*d*): δ (ppm) 165.6 (C = O), 147.6 (C), 142.5 (CH), 137.8 (CH), 130.0 (C), 125.9 (CH), 123.4 (CH), 120.4 (CH), 108.8 (CH-Aromatic), 107.7 (CH-Aromatic), 101.0, 45.5 (CH_2_), 43.4 (CH_2_), 26.7 (CH_2_), 25.7 (CH_2_), 24.5 (CH_2_). HRMS (ESI) calc. for C_17_H_19_NO_3_ [M+H]^+^: 285.1364, found: 285.1362 [26,27].

### 2.4. Fungal Strains and Growth Conditions

Here, a reference fungal strain, *C. albicans* SC5314 was employed. Additionally, a fluconazole (FLC)-resistant isolate (CAAL256) that was previously characterized and identified in our group [28] was included to evaluate the effect and mechanisms of action of piperine against *C. albicans. C. albicans* and non-*albicans Candida* clinical isolates were employed to determine the minimal inhibitory concentration (MIC). The *C. albicans* mutant strains (heterozygous deletion) from the *Candida albicans* library were included to help understand the reduction in virulence, *efg1Δ*/*EFG1*, *cph2Δ*/*CPH2, hwp2Δ*/*HWP2, ume6Δ*/*UME6, flo8Δ*/*FLO8, tup1Δ*/*TUP1* and *nrg1Δ*/*NRG1,* and mechanisms of action of piperine against *C. albicans: ali1Δ*/*ALI1, cox4Δ*/*COX4*, and *hog1∆*/*HOG1* [29]. The mutant strains are detailed in Appendix A. The clinical isolate was provided by San Ignacio University Hospital and the microorganism collection of the Pontificia Javeriana University (Bogotá. Colombia). All cells were stored at 4 °C on YPD agar plates (1% yeast extract, 2% peptone, 2% dextrose, and 2% agar) prior to each experiment.

### 2.5. Antifungal Activity Assay

The antifungal broth microdilution (BMD) method was conducted in accordance with the CLSI BMD-M27-A4 rules, with minor modifications [30,31,32]. First, stock solution (piperine) was prepared at 80 mg/mL in dimethyl sulfoxide (DMSO). *Candida* strains were cultured in RPMI 1640 (RPMI; Gibco, Grand Island, NY, USA) with 2.1 mM L-glutamine and buffered with 165 mM MOPs, pH 7.0) culture medium and adjusted turbidity with 0.5. McFarland Standard. A total of 100 μL of yeast inoculum was added to each well of a 96-well plate containing serial two-fold dilutions of piperine (ranging from 8 to 2048 μg/mL). After incubation at 37 °C for 24 and 48 h, the minimum inhibitory concentration (MIC) was determined according to guidelines of the Clinical and Laboratory Standards Institute (CLSI) [33]. Controls included a positive control in RPMI or DMSO, and a control group treated with fluconazole (FLC) at concentrations ranging from 0.125 to 128 μg/mL using a two-fold serial dilution. Sterility controls comprised Roswell Park Memorial Institute (RPMI) 1640 medium (Gibco, Grand Island, NY, USA) and a saline solution (SS) with a concentration of 0.85%.

The MIC was determined as the lowest concentration of piperine that caused 80% inhibition of visible fungal growth compared to its corresponding positive control (without the extract). MIC values were determined through three independent assays. Following MIC determination, the minimum fungicidal concentration (MFC) was determined by adding an aliquot from each well that showed antifungal activity to Petri dishes containing Sabouraud Dextrose Agar (SDA; Difco) and incubated for 24 h at 37 °C [34].

### 2.6. Time-Kill Kinetic Assay

The culture conditions in this study remained consistent with those in the previously described antifungal activity assay. Briefly, piperine (at concentrations ranging from 8 to 1024 μg/mL) was introduced into a 96-well plate along with a *C. albicans* suspension to achieve a final concentration of 0.5–2.5 × 10^3^ cells/mL in RPMI-1640 medium (with MOPS, pH 7.2). These plates were then loaded into a Bioscreen C equipment, maintained at 37 °C, and subjected to 24 h of continuous agitation. Measurements were taken at one-hour intervals, recording the optical density at 600 nm. The data collected were subsequently analyzed using Bioscreen software (Growth Curves USA, Piscataway, NJ, USA) at specified time points, and the experiments were conducted in triplicate [35,36].

### 2.7. Hyphal Morphogenesis of Candida albicans

To assess the impact of piperine on hyphal morphogenesis during yeast-form growth in *C. albicans*, RPMI 1640 and Yeast–Peptone–Dextrose (YPD) medium plus 10% fetal bovine serum (FBS) were employed according to established procedures [15,37]. *C. albicans* strains were cultured overnight in 2 mL Sabouraud dextrose broth (SAB) at 30 °C with shaking. Overnight cells were collected by centrifugation at 5000× *g* for 10 min. Following supernatant removal, cells were subjected to two washes with phosphate-buffered saline (PBS) at pH 7.2 and adjusted to an inoculum of 1–5 × 10^6^ cells/mL. Afterwards, cultures were incubated in RPMI 1640 medium with shaking (220 rpm) for 4 h at 37 °C with specified concentration of piperine, while RPMI transition-inducing media was employed as a control. The formation of hyphae was examined using an inverted microscope (ICX41) equipped with an OD400UHW-P digital microscope camera at ×40 magnification. To calculate the percentage of hyphae formed, we divided the number of hyphae formed by the total number of *C. albicans* cells.

The yeast and hyphal morphogenetic transition states were also induced on solid media by using YPD agar plates (10 g yeast extract, 20 g peptone, 2% dextrose and 10% FBS) [38] and Spider medium (1% brain heart infusion broth, 1% mannitol, 0.2% dibasic potassium phosphate, pH 7.2) [39] with or without different concentrations of piperine. The cells were incubated for 7 days at 37 °C, and images were captured using differential-interference-contrast microscopy (*n* = 3).

### 2.8. Formation of C. albicans Biofilm

To establish biofilms, *C. albicans* strains were examined in 96-well microplates with minor adjustments [40]. Standardized cell suspensions (100 μL fungal suspensions containing 1.0 × 10^6^ cells/mL in RPMI 1640 buffered with MOPS) were seeded into 96-well plates with or without piperine (8–2048 μg/mL). The non-adherent cells were removed after a 1 h adherence incubation, 200 μL fresh RPMI medium was applied to each well, and the incubation was continued for another 24 h at 37 °C. After washing twice with phosphate-buffered Saline (PBS), the wells were filled with 200 µL of RPMI-1640 supplemented with 10 µL of 700 µM resazurin (Sigma-Aldrich, St. Louis, MO, USA) and incubated at 37 °C for 2 h in the dark (*n* = 3). The fluorescence intensity was measured every hour, and it was quantified at λ excitation  =  560 nm/λ emission  =  590 nm using a microplate reader (Model 550 Microplate Reader, Bio-Rad, Milan, Italy). Caspofungin (CAS) was employed as a standard antifungal drug at concentrations ranging from 0.03 to 1 μg/mL.

### 2.9. Scanning Electron Microscopy (SEM)

For SEM, *C. albicans* cells (2 mL of a suspension containing 1 × 10^6^ cells/mL) were pretreated for 4 h with piperine at different concentrations and incubated a 37 °C. The pellet was washed with sterile PBS and fixed in a mixture of aldehydes (2% paraformaldehyde, 2.5% glutaraldehyde in 0.05 M sodium cacodylate buffer, pH 7.2) and left at room temperature for 18 h. Following this, a series of three washes with 95% ethanol, with 10 min intervals between each wash, were performed, followed by three washes with 100% ethanol, with 20 min intervals between each wash. Subsequently, the samples were preserved in 100% ethanol until the analysis [41,42]. After the samples were completely dry, they were scrutinized using a scanning electron microscope using a Tescan Lyra 3 (TESCAN, Brno, Czech Republic) at Andes University.

### 2.10. Membrane Permeabilization Assay

Propidium iodide (PI) staining was carried out to evaluate the cell death induce by piperine [43]. *C. albicans* cells (1 × 10^7^ cells/mL) in YPD medium was treated with specified piperine concentrations (64, 128 and 256 µg/mL) for 4 h at 37 °C. Cells were washed twice with PBS. Then, a solution containing 5 µL of calcofluor white (CW) (5 µg/mL) and propidium iodide (PI) (5 µg/mL) was added to the cell pellet, and the volume was adjusted to 500 µL with PBS. The staining mixture was incubated for 10 min at 4 °C in the dark, followed by two PBS washes. Clotrimazole (5 µg/mL) was used as positive control. The stained samples were visualized using a LEICA DMi8 microscope (Leica Microsystems, Madrid, Spain).

### 2.11. Measurement of Mitochondrial Membrane Potential

Rhodamine 123 staining (Sigma-Aldrich), as previously described with minor adjustments, was carried out. [44]. *C. albicans* cells (1 × 10^7^ cells/mL were exposed to various concentrations of piperine for 2 h at 35 °C. After treatment, the cells underwent two washes with PBS and were then resuspended in a solution containing 25 µM rhodamine 123. This suspension was incubated at 35 °C for 20 min. After washing three times, the fluorescence intensity was measured using a Varioskan spectrofluorometer LUX (Thermo Scientific™, Waltham, MA, USA) at λ excitation  =  480 nm/λ emission  =  530 nm. Cells without treatment were used as negative controls. As positive controls, we used sodium azide (NaN_3_) (5 mM) for 20 min.

### 2.12. ROS Detection

Intracellular ROS levels were assessed using DCFH-DA (2′,7′-dichlorofluorescein diacetate) (Merck-Millipore, Darmstadt, Germany), following established methods [35,44]. Briefly, *C. albicans* cells (1 × 10^7^ cells/mL) were exposed to various concentrations of piperine for 1 h. After treatment, the cells were rinsed with PBS, and cells were incubated with 20 µg/mL DCFH-DA, at 35 °C for 30 mim. The fluorescence intensity was measured using a Varioskan spectrofluorometer LUX, and detected with excitation/emission at 485 nm/530 nm. For the purpose of investigating the connection between ROS generation and the effectiveness of piperine against *C. albicans*, the ROS scavenger N-Acetyl-L-cystein (NAC) was employed. The experimental procedure involved suspending 1 × 10^7^ cells/mL in 12.5 mM sodium acetate with 60 mM NAC for 30 min at 37 °C. The cells were collected, washed with PBS, treated with various piperine concentrations, and incubated for 1 h and 30 min at 37 °C. Amphotericin B (AmB) at concentrations of 4 μg/mL and 64 μg/mL was utilized as a positive control for SC5314 and CAAL256, respectively.

### 2.13. Checkerboard Assay

A checkerboard test was conducted with some modifications. Briefly, piperine was tested against FLC (concentrations: 0, 0.06, 0.12, 0.25, 0.50, 1, and 2 times the MIC), and piperine was also tested against NAC (concentrations: 0, 1.56, 3.12, 6.25, 12.5, 25, and 2 times the MIC). The mixtures were prepared, and the fungal inoculum was added to each well (final concentration of 0.5–2.5 × 10^3^ cells/mL) and incubated for 48 h at 37 °C, and the new MIC values were determined. The fractional inhibitory concentration index (FICI) was determined using the formula FICI = [A in combination/MICA alone] + [P in combination/MICP alone], where MIC, MICA, and MICP are the MICs of the individual piperine, NAC, and/or antifungal. A-and-represents the MICs when used in combination. Antifungal interactions were classified as: (1) synergistic effects FICI ≤ 0.5; (2) additive effects 0.5 < FICI ≤ 1; (3) indifferent 1 < FICI < 4; and (4) antagonistic effects FICI ≥ 4.0 [35,45].

### 2.14. Hemolytic Activity Assay

The hemolytic potential of piperine was evaluated using the previously described method [43]. 5 mL of blood was collected from a healthy volunteer, laced into Ethylenediamine tetraacetic acid (EDTA) vials (10%), and then centrifuged at 1000× *g* for 5 min. The erythrocyte-rich fraction was resuspended in PBS and subjected to two rounds of centrifugation at 1000× *g* for 7 min. A total of 100 μL of piperine (in concentrations ranging from 4 to 2048 μg/mL) was added to 100 μL of erythrocytes (hematocrit of 4%) in U-bottomed 96-well plates. The mixture was incubated at 37 °C for 2 h. The reaction was stopped by centrifugation at 500× *g* for 10 min. Subsequently, 100 μL of the supernatant was transferred in a 96-well flat-bottomed, and the hemolytic activity was measured at 450 nm. Negative control: PBS. Positive control: Tween-20 (20% *v*/*v*) in PBS (*n* = 3).

### 2.15. Cell Viability Assay

Cell viability was assessed using the 3-(4,5-dimethylthiazol-2-yl)-2,5-diphenyltetrazolium bromide (MTT) assay in a primary culture of human foreskin fibroblasts, with minor modifications [46,47]. Cells were added to a 96-well plate at 2.5 × 10^4^ cells per well and allowed to adhere and proliferate for 24 h in DMEM (Dulbecco’s Modified Eagle’s) medium at 37 °C under a 5% CO_2_. Subsequently, piperine solutions were prepared at concentrations ranging from 8 to 1024 μg/mL using DMEM medium as a solvent (without 10% Fetal Bovine Serum). The prepared concentrations of piperine were subsequently added to wells containing adhered cells. Cells were incubated at 37 °C in a 5% CO_2_ condition for 24 h. After this time, the medium was removed, cells were washed with PBS, and 30 μL of MTT (1 mg/L in PBS) was added to each well-plate. The plates were then incubated at 37 °C for 4 h in a 5% CO_2_. Following incubation, the MTT solution was discarded, and to solubilize the formazan crystals generated from MTT metabolism, 100 μL of DMSO was added. The microplate reader BioTek ELx800 absorbance reader (Agilent, Santa Clara, CA, USA) was used to calculate cell viability and the optical density (OD) at 495 was measured. An incomplete DMEM culture medium containing 10% MTT was employed as a negative control. Furthermore, IC_50_ (half-maximal inhibitory concentration) was computed by graphing viability against the logarithm of the concentration.

### 2.16. Statistical Analysis

The experiments were carried out with a minimum of three biological replicates and at least two technical replicates. Data are presented as means ± standard deviation (SD) and were analyzed by ANOVA and Dunn’s multiple comparison test. *p* < 0.05 was considered significant. The statistical models were achieved using GraphPad software (version 7) (GraphPad Software Inc., La Jolla, CA, USA).

## 3. Results and Discussion

### 3.1. Structural Elucidation of Isolated Piperine

Compound **1** was obtained as a yellow crystalline solid and initially characterized on TLC by the development of an orange color when treated with Dragendorff’s reagent. The melting point ranging between 127 and 129 °C. The NMR spectra of compound **1** (Appendix A) exhibited characteristic signals observed for alkaloid-type compounds, specifically reported for the species *P. nigrum*.

The NMR analysis confirmed the presence of aromatic signals, as evidenced by the ^1^H NMR signals at δ_H_ 6.94–6.92 (m, 2H) and the Distortionless Enhancement by Polarization Transfer including the Detection of Quaternary Nuclei (DEPTQ) signals at δ_C_ 147.6 (C), 130.0 (C), 123.4 (CH), and 107.7 (CH). Additionally, a characteristic signal at δ_H_ 5.94 (s, 1H) indicated the presence of a methylenedioxy group attached to the aromatic ring, supported by the signals in DEPTQ at δ_C_ 147.6 (C). Furthermore, specific signals at δ_H_ 3.60–3.42 (m, 4H), 1.67–1.59 (m, 4H), and 1.56–1.48 (m, 2H) confirmed the presence of a piperidine ring, as was also indicated by the signals in DEPTQ at δ_C_ 45.5 (CH_2_), 43.4 (CH_2_), 26.7 (CH_2_), 25.7 (CH_2_), and 24.5 (CH_2_). Additionally, a carbonyl group was observed in DEPTQ at δ_C_ 165.6. The two-dimensional NMR spectra were analyzed, and through comparison of the spectroscopic data with the existing literature, compound **1** was identified as piperine. Finally, the purity of the isolated piperine was confirmed to be greater than 95% through an analysis carried out by UPLC-DAD.

### 3.2. Piperine Inhibit C. albicans Growth

First, Piperine was found to be a growth inhibitory agent against *Candida* spp. It showed a moderately active MIC_80_ [48], with MIC values of 1024 μg/mL for the *C. albicans* strain SC5314 (FLC-sensitive) and 512 μg/mL for the *C. albicans* strain CAAL256 (FLC-resistant) at 24 h. The growth inhibitory effect of piperine was also evident with non-*albicans* species, for example, *Candida parapsilosis* and *Candida tropicalis*. MFC values of 2048 μg/mL were observed for all strains except *C. albicans* SC5314 and *Candida krusei*, which exceeded this concentration after 24 h (Table 1). It is worth noting that the antifungal activity persisted consistently for up to 48 h for CAAL256. This is an interesting finding, considering the emergence of fluconazole-resistant *C. albicans* strains [49].

To date, only one documented study has investigated the antifungal effects of piperine naturally isolated from *P. nigrum*. This study disclosed an MIC value of 3125 μg/mL against *C. albicans* [18]. Nevertheless, other synthetic piperine research has yielded conflicting results. Thakre et al. indicated a strong antifungal activity with MIC values between 2.5 and 15 μg/mL against resistant isolates and strains of *C. albicans*, differentially susceptible to FLC, using a broth microdilution assay [22], whereas Mgbeahuruike et al. exhibited that the commercial piperamide compound piperine was active against *C. parapsilosis* (ATCC 7330), with a MIC value of 39 µg/mL, and presented MIC values of 78 µg/mL against *C. albicans* (ATCC 10231), *C. glabrata* (ATCC 2001), and *C tropicalis* (ATCC 750) using the same method [50]. One study using the agar well diffusion method showed that commercial piperine (at 1 mg/mL) was effective against *C. albicans* (ATCC 14053), while it showed the least antifungal activity against *C. parapsilosis* (ATCC 22019), *C. rugosa* (ATCC 10571), and *C. krusei* (ATCC 14243) [5]. Conversely, other research has reported no activity against *C. albicans* (ATCC 90028) even at the maximum concentration examined, which was 1024 μg/mL [21].

In this study, the activity of piperine was comparatively lower when contrasted with studies utilizing commercially available piperine. This discrepancy suggests the possibility of minor distinctions in the sensitivity and accuracy of the method used [51] in contrast to synthetic piperine from pharmaceutical companies, which undergoes meticulous quality control during the manufacturing process [52]. Nevertheless, the purification method employed in this study is distinguished by its speed and reproducibility, yielding a considerable amount of reasonably pure piperine in a notably rapid and cost-effective manner [25,53]. Additionally, it exhibits the potential for its incorporation into drug formulations as an active pharmaceutical ingredient (API). Certainly, despite the abundance of accessible synthetic drug compounds, many pharmaceutical companies are now focusing on the development of plant-derived APIs to avoid cost and complexity [54].

Regarding the kinetic results, the treatment of 8–1024 μg/mL of piperine inhibited the growth of the yeasts by ~80% in both *C. albicans* SC5314 and CAAL256 until 13 h but showed growth after 24 h, indicating a fungistatic activity (Figure 1a,b). Still, at the maximum tested concentration (2048 μg/mL), piperine demonstrates a fungicidal effect on both strains. Limited understanding exists regarding the mechanism underlying the antifungal activity of piperine. However, findings from an apoptosis assay suggest that it demonstrates fungicidal activity at elevated concentrations against both sensitive (ATCC 90028) and resistant (ATCC 10231) strains of *C. albicans* equally in our study [22].

Since *C. albicans* is the most common species associated with candidiasis (65.3%) and the most frequently isolated [55], the study has focused on investigating the role of a plant-derived alkaloid molecule, against both FLC-sensitive and FLC-resistant strains of *C. albicans*. This approach enabled us to explore virulence factors as plausible pharmacological targets and to delve deeper into the mechanism of action associated with a natural compound, as illustrated below.

### 3.3. Piperine Inhibits Biofilm Formation and Hyphal Morphogenesis of C. albicans

*C. albicans* is the most common cause of invasive candidiasis on a global scale [56]. Robust epidemiologic evidence shows a clear association between biofilm formation and mortality from invasive candidiasis. There is also evidence that *C. albicans* has a greater propensity to form biofilms than other *Candida* species and that biofilms are up to 1000 times more resistant to azoles than planktonic cells [57]. Given the increasing drug resistance of *C. albicans* biofilms, there is an urgent need to identify alternative sources of antibiofilm agents for the treatment options for biofilm infections.

Recently, the ethanolic extract of *P. nigrum* was explored in our group as a promising potential antibiofilm against *Candida* spp. [15], and the results suggested that piperine might play a role in attenuating certain virulence factors in *Candida* spp. Initially, the biofilm inhibitory potential of piperine was evaluated against *C. albicans* strains after 24 h of treatment. Piperine dose-dependent inhibited the biofilm formation of strain SC5314 at concentrations of 32, 64, 128, 256, 512, and 1024 μg/mL by about 54%, 75%, 83%, 95%, and 95%, respectively (Figure 2a). Subsequently, the strain CAAL256 at concentrations of 128, 256, 512, and 1024 μg/mL percentage inhibition of biofilm formation were about 45%, 57%, 70%, and 81%, respectively (*p* ≤ 0.05) (Figure 2b).

These findings align with previous research on piperine, which demonstrated more than 90% inhibition of biofilm in C. albicans 90028 (ATCC) at a concentration of 32 µg/mL [21]. Additionally, piperine remarkably enhanced the inhibitory activity against biofilm cells of both FLC-sensitive and FLC-resistant strains of *C. albicans* compared to the ethanolic extract of *P. nigrum* [15]. This difference could be attributed to the characteristic of the chemical composition of both compounds. Piperine is effective in interacting with biofilm formation because its structure exerts a specific action on the fungus, avoiding the interference of other compounds. Moreover, at the structural level, it was found that this molecule promotes a rapid interaction owing to its lipophilic character, facilitating the alteration of permeability in cell membranes [58,59]. In addition, its potential to inhibit biofilm allows it to weaken the protective barrier, thereby exerting antifungal potential.

Given that piperine can evidently deploy a substantial inhibitory effect on biofilm formation at lower concentrations in the SC5314 strain in comparison to the CAAL256 strain of *C. albicans*, it is suggested that this disparity could potentially arise from distinct metabolic processes within the biofilm of each species [60]. In tune with this, it is speculated that the subtle variation in the FLC-resistant strain could happen due to the expression of resistance genes, particularly those encoding efflux pumps; the link to defects in the enzyme Erg11; or changes in its biosynthetic pathway that affected membrane fluidity and permeability [8]. Hence, controlling the early stages of biofilm formation emerges can be a prevailing strategy to control infection and its dissemination [61].

The yeast-to-hyphal conversion constitutes another crucial virulence trait of *C. albicans* and represents a prerequisite for both biofilm formation and the subsequent invasive growth of *C. albicans*, enabling the penetration of host tissues and the establishment of infections [62]. Piperine tested under hyphal-inducing conditions significantly reduced hyphal and pseudohyphal formation in *C. albicans*. Based on the results, the transition to hyphae was significantly reduced in SC5314 strain treated with piperine (4–256 µg/mL) in a dose-dependent manner. Hyphal induction was observed by 52.7% in SC5314 only at 4 µg/mL compared to ~90% in the control (Figure 3a). In contrast, the strain CAAL256 showed pseudohyphae induction with more elongated cellular compartments, forming branching chains and few true hyphae, which is consistent with our previous publication in which a low number of cells developed true hyphae in the resistant strain [15]. Incubation with 16 µg/mL resulted in the significant suppression of pseudohyphae by 28.7% compared to ~50% at control, and a dose-dependent response was also evidenced (16 to 256 µg/m) (Figure 3b). These findings are in agreement with those of Priya et al. who reported that piperine exhibits anti-hyphal activities against *C. albicans* ATCC (90028) in a dose-dependent response manner (8 to 128 µg/m) [21], as well as with Thakre et al. who reported the complete inhibition of *C. albicans* ATCC (10231) at 5 µg/mL, while this increasing concentration decreased pseudohyphae formation [22]. Remarkably, it is suggested that piperine effectively inhibits the biofilm and hyphae formation *in C. albicans* by regulating genes associated with biofilm, hyphae, and adhesion. [61,63]. The disruption of hyphal formation in *C. albicans* by piperine, even in the resistant strain, was an anticipated outcome given the observed anti-filament activity of ethanolic *P. nigrum* extract. This expectation arose from the fact that piperine was the major constituent of this extract [15]. However, it is notable that this bioactive compound could be even more promising considering its ability to affect virulence traits of *C. albicans* in lower concentrations.

The above results were correlated with a potential impact of piperine on the hypha formation of *C. albicans* on solid medium. In both the SC5314 and CAAL256 strains, filamentation was not observed on the surface and edges with YPD containing FBS supplemented by treatment with 32 and 64 μg/mL of piperine, respectively, whilst the colonies were markedly smoother than in the respective control—grown in absence of piperine—which showed a wrinkled phenotype when grown in Spider medium (Figure 3c). Similar to filamentous morphology, wrinkling in *C. albicans* is regarded as a virulent trait, as it requires hyphal development and adhesion production in the formation of wrinkled colonies [64]. A previous study carried out with piperine showed that even at the lowest concentration (8 μg/mL), piperine inhibited the wrinkling of *C. albicans* cells [52]. The prevention of virulent morphologies associated with invasion and biofilm formation then provides evidence that piperine could have the potential to effectively hinder invasive candidiasis [21].

For a deeper examination of the effect of piperine on hyphal transition, the cells were examined by SEM. Results revealed that compared to control yeast, piperine at 8 to 64 µg/mL generates alteration on the surfaces of treated cells, with the observation of pores and protuberances in the cells (Figure 4). After exposition to both strains for 4 h to 64 µg/mL, it was observed that compared to untreated cells, this concentration generates higher alterations in yeast wall and indentions in the form of yeast. Also, at 8 µg/mL, cracks in the cell wall and blisters can be observed in the appearance of the hyphae. It is hypothesized that when the strains are grown in the presence of this natural metabolite, bud and hyphal cells exhibit alterations in cellular structure, implying a possible action of piperine on *C. albicans* cell wall, as well as on arrest cell cycle in G2 m phase [22]. This is an encouraging result since the study was examined in both the FLC-sensitive and the FLC-resistant strains, laying the foundations for the investigation of different phenotypes and responses in strains with growth potential under stressful conditions. Nonetheless, additional studies are required to ascertain the full spectrum of activity.

To support the results in which the yeast-to-hyphal transition is a critical morphogenetic feature during pathogenesis, some hyphal-specific mutant genes of *C. albicans* were tested under both hyphal-inducing conditions and in the presence of piperine. Transcriptional regulators of filamentous growth, such as *EFG1, CPH2, UME6* and *FLO8*, transcriptional repressor that negatively control filamentous as *TUP1* and *NRG1* (mutants display a high degree of filamentous growth or hyperfilamentation [65]), and hyphal specific gene, as *HWP2* were evaluated to assess the requirement for these genes in the piperine response.

Figure 5 shows that under hyphal induction, nearly all *efg1Δ*/*EFG1*, *cph2Δ*/*CPH2, hwp2Δ*/*HWP2, ume6Δ*/*UME6, flo8Δ*/*FLO8, tup1Δ*/*TUP1* and *nrg1Δ*/*NRG1* cells exhibited a reduction in the percentage of filamentation of untreated cells compared to the wild type (WT), and more pseudohyphae and clusters of cells were regularly observed in almost all mutant strains tested.

Notably, piperine at the lowest concentration did not affect yeast-to-hyphal transition in *EFG1* mutant cells. Conversely, the results demonstrated a significant reduction in the transition to germ tubes in *CPH2* and *HWP2* strains treated with piperine from 8 to 128 µg/mL in a dose-dependent manner after 4 h of incubation. Furthermore, it was observed that *UME6* and *FLO8* mutants exhibited enhanced pseudohyphal growth, even in the presence of piperine compared to the WT. Previously, authors reported that mutants lacking UME6 exhibit abnormal germ tube formation and an inability to sustain hyphal growth, leading to the production of elongated yeast cells or short pseudohyphae [66]. Interestingly, Carlisle et al. revealed that when Ume6 levels are low, a subset of filament-specific genes is activated, resulting in the majority of cells growing as pseudohyphae [67]. Cao et al. conducted a study on the deletion of both *FLO8* alleles, resulting in mutants that displayed a complete inability to respond to hyphal induction in various liquid hyphal-inducing conditions. Nonetheless, when the flo8/flo8 mutant was transformed with a wild-type (WT) copy of FLO8 under its native promoter, partial defects in hyphal morphogenesis were observed. Additionally, the *FLO8*/*flo8* heterozygote exhibited reduced expression of hypha-specific genes [68]. Then, it is suggested that piperine treatment may affect either the morphology or the expression of UME6 and FLO8 as the concentration of piperine increases, and the pseudohyphal form is largely retained.

On the other hand, the deletion of *TUP1* in the presence of piperine led to a reduction in hyphal growth. On the contrary, pseudohyphal growth was maintained at different concentrations of piperine. Despite the fact that the results were apparently the same as in the WT strain, it was observed that *C. albicans tup1Δ*/*TUP1* cells constitutively form long, non-aggregating pseudohyphal filaments [69] (Appendix A). Finally, *NRG1* treated with piperine at the lowest concentration significantly affected the percentage of hyphal transition in a manner similar to that of SC5314, but elongated filaments were discerned (Appendix A). Braun et al. reported that Nrg1 is known to repress filamentous growth in *Candida*, likely by interacting with the co-repressor Tup1. In the absence of *nrg1*, mutant cells tend to exhibit a predominantly filamentous phenotype even under conditions that do not typically induce filamentation. Additionally, their colony morphology is reminiscent of that seen in *tup1* mutants [65].

The regulation of morphogenesis in *C. albicans* is a complex process. However, to date, all the evidence aims to a limited set of regulatory molecules where all signals seem to converge *CPH1, CPH2, TEC1, RIM101, EFG1, NRG1,* and *UME6* [70,71]. Here, the heterozygous *EFG1, CPH2*, *HWP2, UME6, FLO8,* and *NGR1* mutants showed a propensity for filamentation in the absence of piperine (Appendix A). In the context of other reports, it appears that the homozygous deletion mutant *efg1ΔΔ* is required to reduce in vitro and in vivo filamentation in *C. albicans* [72]. Thus, it is suggested that the presence of a copy of the WT, driven by its own promoter, corrects the morphological alterations. On the other hand, the homozygous mutant *cph2ΔΔ* strains did not show any noticeable defects in germ tube or hyphal development in various liquid media that typically induce hyphal growth, including serum, suggesting that Cph2 could potentially play a role in mediating medium-specific signals in hyphal development [73]. Additionally, Hayek et al. previously showed that Hwp2 does not seem to play an important role in filamentation in liquid media, although the mutant strain did show a minor reduction in hyphal formation in the presence of serum. However, a more noticeable difference between the mutant and wild-type strains was observed in solid media, which confirms the influence of glucose on filamentation [74]. Therefore, it is not unexpected that the *hwp2Δ* phenotype exhibited differences compared to the parental strain [75].

The morphogenetic switches induced by piperine were found to be dependent on these regulators. According to a recent report, the gene expression analysis indicated that piperine significantly downregulating the expression some biofilm and hyphal specific genes. These findings were consistent with the results of phenotypic assays (*EFG1, CPH1, UME6, and HWP1*) [21]. Taken together, these results suggest that piperine can inhibit biofilm production by preventing the formation of germ tubes and hyphal development in *C. albicans* mutant genes. Piperine’s effects are not limited to biofilm structure but also extend to the expression of hyphal-associated adhesins, suggesting that this alkaloid targets a pathway essential for hyphal morphogenesis under inductive conditions.

### 3.4. Piperine Affects Membrane Permeability of C. albicans

Notwithstanding several biological activities of piperine having been demonstrated in both preclinical and clinical studies, its underlying mechanism remains unknown [76]. Piperine exhibits a wide range of activities, including antibacterial, antifungal, anticancer, anti-inflammatory, and more [77,78]. However, unlike the antimicrobial functions, the potential mechanisms explaining the effect of piperine against *C. albicans* are less well understood. Having established the inhibitory effects of piperine on the virulence factors of *C. albicans* strains, further exploration was undertaken to understand the underlying mechanisms governing its antifungal efficacy. First, the membrane permeability of *C. albicans* following piperine treatment was assessed by utilizing propidium iodide (PI) staining. PI serves as a nucleic acid marker that cannot penetrate intact cell membranes. Nevertheless, during apoptosis or necrosis, alterations in plasma membrane and nuclear membrane permeability facilitate the entry of PI into cells, leading to the induction of red fluorescence [79]. A dose-dependent membrane permeabilization of *Candida* SC5314 and CAAL256, as indicated by the cellular accumulation of PI, was evident at concentrations of 64 to 256 µg/mL for 4 h (Figure 6). The detection of clotrimazole signals resulted in the observation of red fluorescence localized to intracellular compartments of the cells by its action damaging the permeability barrier in the fungal cytoplasmic membrane [80].

A small proportion of SC5314 cells did have discernible but low levels of PI staining, indicating partial permeability to the dye. However, CAAL256 cells treated with piperine at 256 µg/mL showed more membrane permeabilization compared to SC5314 cells, indicating a considerable defect in the cell plasma membrane. These results agree with previous studies in which piperine and piperidine derivatives containing a triazole moiety disrupt membrane integrity, leading to oxidative stress followed by cell cycle arrest and subsequent death by apoptosis in *C. albicans* [22] and in *C. auris* [76], respectively. The results of the time–kill assays indicate that piperine has fungistatic properties at low concentrations. Seemingly, the metabolite has a moderate antifungal activity and promotes apoptosis during low concentration exposures, which is likely to damage the membrane permeability barrier of *C. albicans* and to cause cell rupture or dysfunction. According to Davey et al., within a range of pertinent stress conditions, the membranes of yeast species have the potential to undergo reversible permeability to PI [81]. Nonetheless, it is impossible to rule out the possibility that at least some of these cells may be on their way to cell death. Therefore, complementary research is needed to determine the level of damage that the yeast cell membrane can sustain. In summary, the modification of membrane properties induced by piperine may be influenced by its nonpolar nature through interaction or association with lipids and hydrophobic parts in the vicinity of proteins [82].

### 3.5. Piperine Increases Intracellular Concentration of ROS and Causes Disruption of Mitochondrial Membrane Potential

To ascertain the subsequent events following piperine uptake into fungal cells and to determine the primary cause of fungal membrane damage, attention was focused on intracellular ROS levels, which were monitored using the H_2_DCFDA fluorescent dye. Piperine significantly increased the H_2_DCFDA fluorescence intensity after treatment with piperine compared to the control. The results showed that piperine can induce ROS generation in both strain SC5314 and strain CAAL256, but the MFI of FLC-resistant cells treated with piperine was still significantly increased (32, 64, 128, and 256 μg/mL). The results above indicate that the mechanisms underlying piperine in the FLC-resistant strain differ from those mediating the FLC-sensitive strain. We suggest that while the absence of ergosterol in CAAL256 has not been specifically linked to FLC-resistance, mutations in ergosterol biosynthesis genes may lead to abnormal membrane structure and functionality [83]. These mutations could consequently affect tolerance to oxidative stress.

Piperine induced ROS generation by increasing reactive oxygen species that activated programmed cell death in *C. albicans*, which is consistent with previous studies [22]. According to Thakre et al., proteomic data suggest enhanced oxidative stress in response to piperine, as evidenced by the upregulation of four proteins (energy generation and mitochondrial proteins) associated with oxidative stress [22]. Simultaneous treatment with piperine concentrations and NAC resulted in a decrease in ROS levels. NAC partially reversed the generation of total and mitochondrial-derived ROS in response to piperine exposure (Figure 7a,b). One plausible interpretation of this finding could be that the antioxidant role of NAC in *C. albicans* is not solely based on direct ROS scavenging [84]. Rather than primarily focusing on a substantial reduction in their quantity, it is conceivable that the emphasis lies in mitigating the deleterious effects of ROS.

Considering the fact that *C. albicans* Hog1 contributes in the cell response to osmotic and oxidative stresses [85], the possible contribution of Hog1 in the response of strain SC5314 to piperine was determined. The results showed that the *hog1∆*/*HOG1* mutant decreased the intensity of fluorescence, compared to the WT, yet ROS generation in this strain started to increase upon exposure to piperine at 256 μg/mL (Figure 7c). The results also displayed that *hog1∆*/*HOG1* mutant was more tolerant to piperine than its wild type. The HOG1 pathway has been proved to play a critical role in the cellular response to oxidative stress. Mutants lacking both copies of the HOG1 gene (*hog1∆*/*hog1∆*) reveal increased sensitivity to oxidative agents, indicating the importance of the pathway in protecting cells from oxidative damage [86,87]. However, when examining heterozygous mutants, the presence of just one functional copy of the HOG1 gene is sufficient to mount an effective stress response. Recently, Chang et al. indicated that *C. albicans hog1Δ*/*hog1Δ* is insensitive to ROS-generating peptides [44]. These findings highlight the complex regulatory mechanisms underlying the HOG1 pathway and its importance in safeguarding cells from oxidative stress-induced damage.

Cellular metabolism relies on a constant supply of Adenosine 5′-triphosphate (ATP) from the mitochondria, so any impairment of the respiratory chain function is potentially detrimental to cell viability. Mitochondria play a critical role as the primary intracellular sources of ROS [88]. At the same time, they are susceptible to the deleterious effects of excessive ROS production, leading to disturbances in electron transport, ΔΨm (mitochondrial membrane potential), and ATP generation, making them a critical source of ROS as well [89,90]. As the overproduction of ROS in mitochondria leads to mitochondrial dysfunction, we next evaluated the decrease in mitochondrial membrane potential of *C. albicans* cells after piperine exposure using the fluorescent dye, rhodamine 123 (Rho 123), to monitor mitochondrial activity, where an increase in fluorescence intensity is expected when there is depolarization of the mitochondrial membrane.

As shown in Figure 8a,b, the MFI (mean fluorescence intensity) of Rho 123 was increased in both SC5314 and CAAL256 cells treated with piperine or sodium azide (used as positive control) compared to untreated control cells. This result indicated that piperine repressed the growth of both strains by abnormally altering the mitochondrial membrane potential. Apparently, the FLC-resistant (CAAL256) strain showed a greater effect on mitochondrial activity at 128 μg/mL piperine. The effect of piperine on mitochondrial membrane redox stability, leading to ROS accumulation, has been reported in only one study, where increased ROS in *C. albicans* subsequently led to apoptosis [22]. It is plausible that the heightened endogenous ROS levels in the FLC-resistant strain stem from mitochondrial dysfunction, potentially resulting in excessive ROS production, perhaps also due to mitochondrial abnormalities. Nonetheless, further investigations are warranted to elucidate these phenomena comprehensively.

Two mutants lacking genes involved in mitochondrial complex I and IV (*ali1Δ*/*ALI1* and *cox4Δ*/*COX4)* were also included in the study. *Ali1* is a membrane-bound NADH-ubiquinone oxidoreductase (mitochondrial respiratory complex I) activity, and *Cox4* is a putative cytochrome c oxidase subunit [91]. The results showed that the *ali1Δ*/*ALI1* and *cox4Δ*/*COX4* mutants treated with concentrations of piperine increased the fluorescence intensity of Rho 123 compared to the WT, a characteristic of mitochondrial dysfunction (Figure 8c,d). Similarly, candidacidal activity showed that *ali1Δ*/*ALI1* and *cox4Δ*/*COX4* cells were resistant to piperine. Remarkably, mutants lacking oxidoreductases displayed increased sensitivity to oxidative agents such as menadione or hydrogen peroxide. These findings imply that these proteins play a crucial role in protecting against the harmful effects caused by reactive oxidative species [92]. It has been reported that agents that disrupt mitochondria can reduce the sterol content in the membrane by interfering with the respiratory chain, thereby lowering intracellular ATP concentrations [93]. Afterwards, ROS accumulation and membrane damage contribute to *C. albicans* cell death.

As proposed in a recent publication involving a natural product, it is credible that piperine initially might stimulate mitochondria to generate additional ROS as an anti-stress response until it surpasses the cellular antioxidant capacity. Subsequently, the excessive ROS levels lead to damage in mitochondrial function and respiratory chain, ultimately contributing to the death of *C. albicans* [94]. In parallel, piperine’s ability to hinder morphological transitions and alter phenotypes while maintaining a low probability of resistance development underscores its therapeutic promise (Figure 9).

### 3.6. Piperine Interact on the Growth of C. albicans with Fluconazole and NAC

First, the SC5314 and CAAL256 strains were used to evaluate the anti-*Candida* actions of piperine with FLC. When piperine was co-incubated with FLC, the combination showed synergistic effect in SC5314 strain. Through the facilitation of piperine, the concentration of FLC exerted MIC value decreased two-fold, while the MIC value of piperine decreased three-fold when combined with fluconazole. Consequently, additivity was obtained against the CAAL256 strain, which decreased the MIC of piperine and FLC by a factor of 5 and 2, respectively (Table 2). Previously, piperine was evaluated in combination with FLC in an FLC-resistant strain of *C. albicans* (ATCC 10231), obtaining a synergistic effect and suggesting the modulation of membrane permeability and signal transduction [22]. Our findings also may indicate that the synergistic effect of piperine and FLC involves an increase in the intracellular levels of ROS in the FLC-sensitive strain (SC5314) of *C. albicans*, since additional studies have reported that the azoles induce the endogenous production of ROS in *Candida* spp. [95].

Since FLC interrupts the conversion of lanosterol to ergosterol, which leads to a disruption in fungal membranes, and ultimately fungal cell death [96], the highest resistance in *C. albicans* CAAL256 towards FLC could be due to the mutation in ERG11, this being a crucial event for the survival of the fungus. This mutation probably altered the protein structure that reduced the binding affinity of fluconazole. On the other hand, if FLC affects membrane permeability in strain SC5314, this might allow piperine to enter the yeast more and result in synergistic activity; likewise—as suggested by Kim et al.—the azole may also induce intracellular ROS production, which could synergistically affect *Candida* growth [35]. A recent assessment explored the synergistic interaction between piperine and phytochemicals like cinnamaldehyde [61] and thymol [63] against *C. albicans*. These studies revealed that the combined application of piperine and these compounds yields greater efficacy compared to individual treatments. This highlights the growing effectiveness of combination therapy in addressing infectious diseases [97].

In addition, the effect of adding NAC to piperine was investigated. In previous studies, NAC has been shown to act as a scavenger of reactive oxygen species (ROS), thereby exerting antioxidant effects [98]. However, our research revealed a notable decrease in the activity of NAC when combined with piperine. Specifically, we observed that only higher concentrations of NAC inhibited colony growth, while an MIC level resulted in the opposite effect when combined with piperine. The FICI method showed indifference, with FICI values of 1.031 for both FLC-sensitive and FLC-resistant strains. The observed result was expected, as it can be attributed to the distinct mechanisms of action exhibited by the two compounds. Consequently, NAC was unable to provide the anticipated protection against oxidative stress induced by piperine. Therefore, it is suggested that the diminished activity of NAC resulted in a significant accumulation of reactive oxygen species (ROS) within the cellular environment generated by the piperine [99].

### 3.7. Piperine Shows No Appreciable Toxicity against Human Red Blood Cells and Fibroblast Cells

In vitro hemolytic activity on human erythrocytes at various concentrations of piperine was examined. None of the concentrations at a range of 4–1024 µg/mL possess any hemolytic activity. At 2048 µg/mL, a lower hemolytic effect of approximately 27% was observed. The results of MTT assay on the viability of the primary cell culture of fibroblasts showed that, piperine at 4–512 µg/mL exhibited percentages of viability above 50%. However, at the highest concentrations (1024 µg/mL), the use of piperine reduced the percentage of cell viability by more than 31%. Based on the calibration curve, the results showed that piperine had an IC_50_ and R^2^ values of 356 µg/mL and 0.6, respectively (Figure 10).

Recently, *P. nigrum* has been demonstrated to be not toxic within a wide spectrum of growth-inhibiting concentrations. Up to 256 µg/mL, the extract exhibited a minor hemolytic effect, and it was not appreciably toxic to Vero cells with cell viability percentages above 85% at concentrations that inhibited virulence factors in *C. albicans* [15], as in our study. Lately, it has been reported that piperine was found to be non-toxic in human buccal epithelial cells (HBECs) and effectively reduced in vivo colonization and increased the survival of *C. elegans* infected with *C. albicans* [21]. Other researchers documented that piperine selectively induced cytotoxicity in HeLa cells, and not in normal cells, because piperine affects cancer cells and senescent cells differently due to different responses of intracellular signaling pathways [100,101]. Therefore, piperine has been revealed to be indirectly effective, although its mechanism of action remains unknown. Finally, on the scale of toxicity calculation, piperine falls within categories IV to VI, with corresponding LD50 ranges from 330 to 12,000 mg/kg, indicating its safety for human consumption. Additional information pertaining to organ compatibility analysis unveils a substantial safety margin of this compound in relation to hepatotoxicity, carcinogenicity, and mutagenicity [102].

## 4. Conclusions

Above all, this study demonstrated that piperine effectively reduced the virulence traits of both FLC-sensitive and FLC-resistant strains of *Candida albicans*. It inhibited biofilm formation, hampered hyphal transition in both liquid and solid mediums, and induced cell distortions as observed in SEM samples. Additionally, piperine exhibited the inhibition of hyphal growth maintenance in heterozygous haploid deletion strains of *C. albicans.* Furthermore, piperine appears to induce the generation of intracellular reactive oxygen species (ROS) and disrupt mitochondrial integrity, ultimately affecting the membrane’s permeability. This alkaloid also demonstrated synergistic antifungal activity with FLC affecting *C. albicans* growth. In terms of toxicity, piperine exhibited no obvious hemolytic effect and showed low cytotoxicity on fibroblast cells at concentrations that inhibited virulence factors in *C. albicans*. Taken together, these findings shed light on the potential of piperine for reducing virulence in *C. albicans* and underscore its promise as a natural alternative for developing new antifungal agents.

## Figures and Tables

**Figure 1 biomolecules-13-01729-f001:**
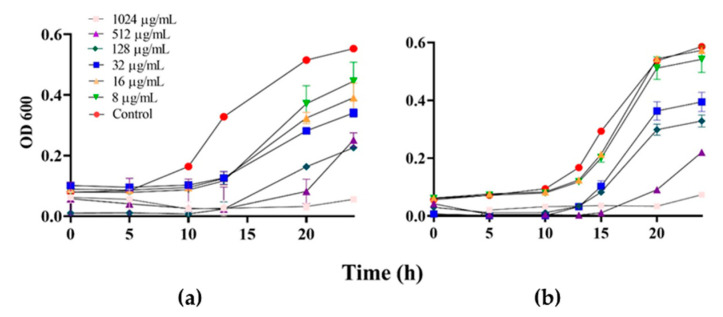
Piperine’s impact on time–kill kinetic curves of *C. albicans* is evident. (**a**) shows the kinetic growth of SC5314, while (**b**) represents the kinetic growth of CAAL256, both treated with piperine concentrations ranging from 8 to 1024 μg/mL at 37 °C for 24 h. Cell growth was monitored via absorbance at 600 nm. These data represent the mean OD ± SD of three independent experiments.

**Figure 2 biomolecules-13-01729-f002:**
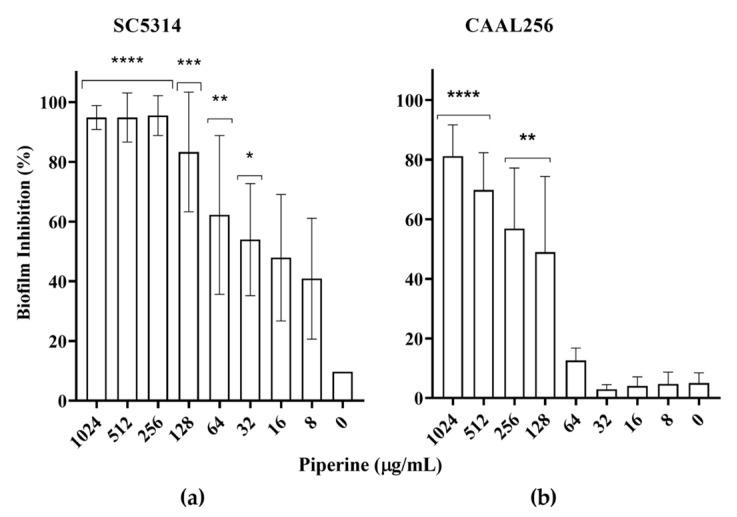
Piperine affects virulence attributes of *C. albicans.* In vitro characterization of the inhibitory activity of piperine on *C. albicans* biofilm formation. (**a**) SC5314; (**b**) CAAL256; *p*-values of < 0.05 were considered statistically significant and denoted as follows: * *p* ≤ 0.05; ** *p* ≤ 0.01; *** *p* ≤ 0.001; **** *p* ≤ 0.0001.

**Figure 3 biomolecules-13-01729-f003:**
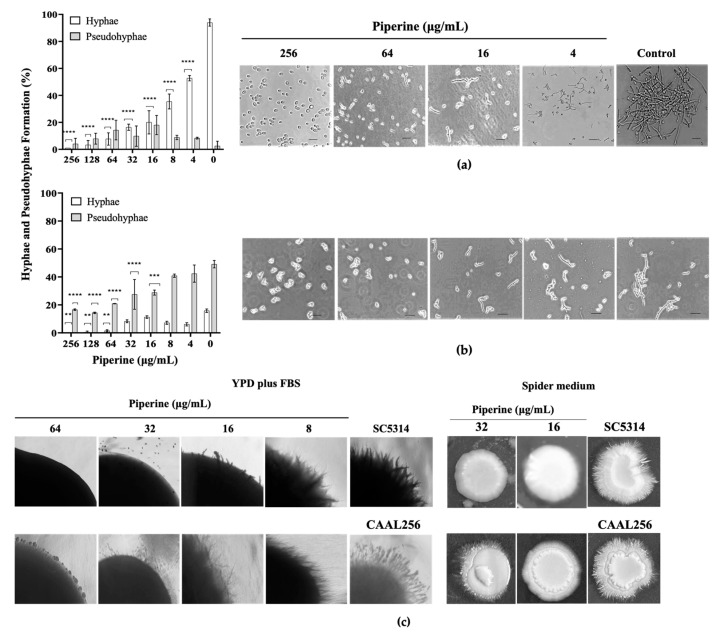
Piperine affects virulence attributes of *C. albicans*. Piperine treatment on yeast-to-hyphal transition in liquid medium. The left panels (**a**,**b**) display the percentage of filamentation in *C. albicans* strains SC5314 and CAAL256, respectively. The right panel includes photographic records. The scale at the bottom right of the images represents 50 µm. Statistical significance was denoted by *p*-values as follows: ** *p* ≤ 0.01; *** *p* ≤ 0.001; **** *p* ≤ 0.0001. (**c**) The effect of piperine on the filamentation of *C. albicans* on solid medium is also presented (bottom panel). This includes images of colony morphology taken after incubation for 7 days at 37 °C in YPD agar plus 10% FBS and Spider medium, both in the absence and presence of piperine. The images were captured at 40× magnification.

**Figure 4 biomolecules-13-01729-f004:**
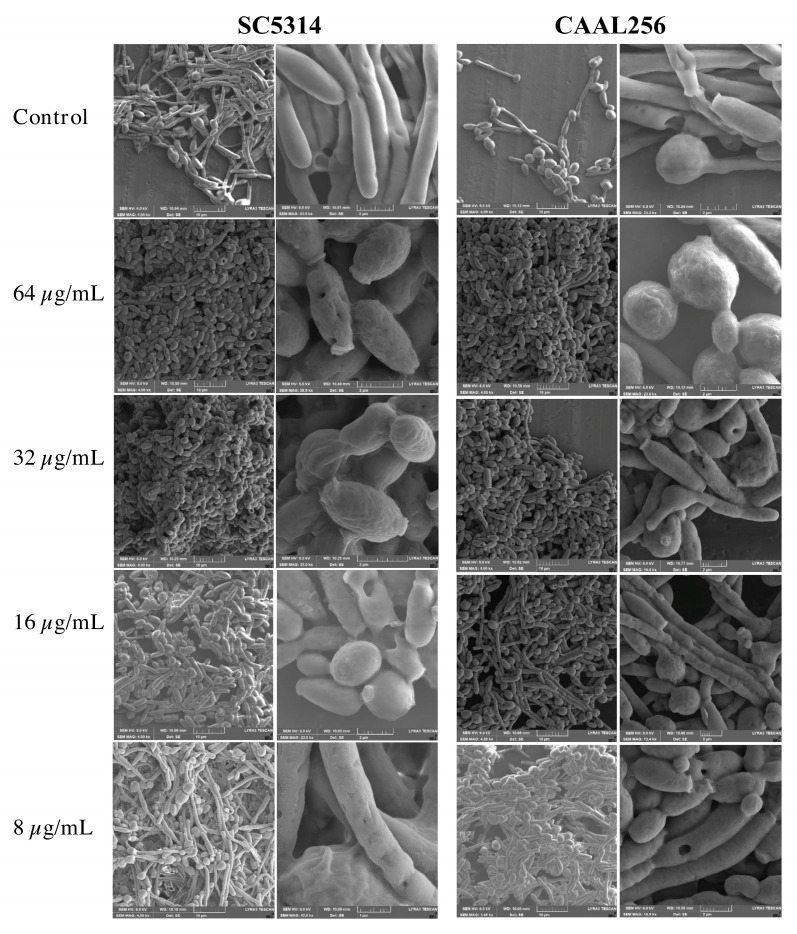
SEM of *C. albicans* after treatment (4 h + supplementation of RPMI-1640 medium) with piperine to 8–64 µg/mL. Control: untreated cells. Image size: column one (left): 10 µm; column two (right): 2 µm.

**Figure 5 biomolecules-13-01729-f005:**
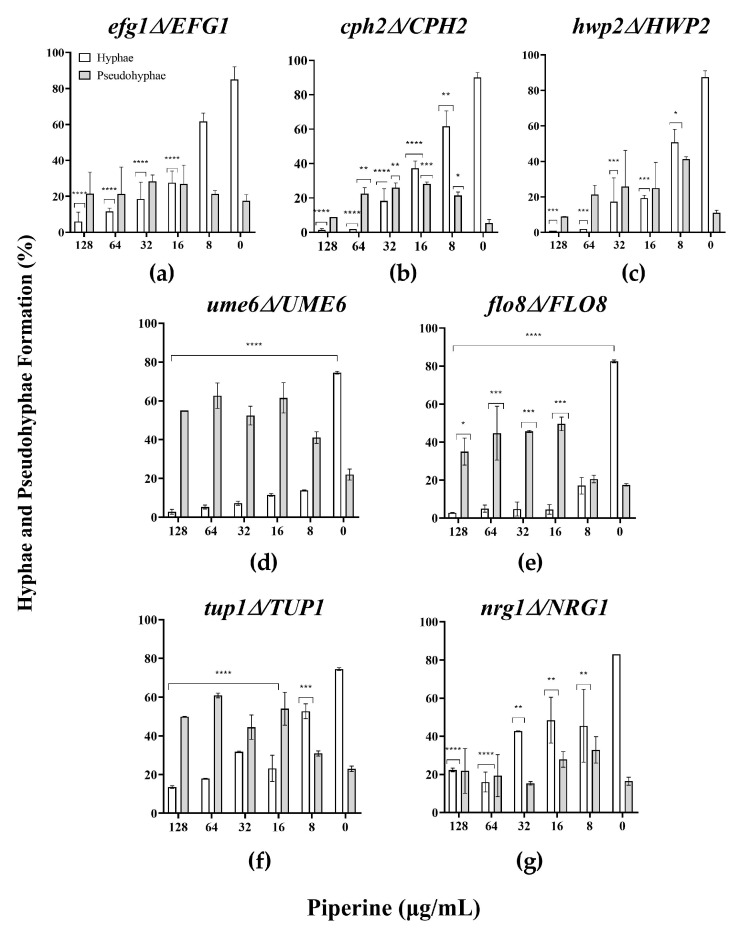
Piperine slightly affects yeast-to-hyphal transition in liquid medium in mutant strains of *C. albicans*. Percentage of filamentation in (**a**) *efg1Δ*/*EFG1*, (**b**) *cph2Δ*/*CPH2,* (**c**) *hwp2Δ*/*HWP2,* (**d**) *ume6Δ*/*UME6,* (**e**) *flo8Δ*/*FLO8,* (**f**) *tup1Δ*/*TUP1,* and (**g**) *nrg1Δ*/*NRG1* strains. *p*-values less than 0.05 were considered statistically significant and denoted as follows: * *p* ≤ 0.05; ** *p* ≤ 0.01; *** *p* ≤ 0.001; **** *p* ≤ 0.0001.

**Figure 6 biomolecules-13-01729-f006:**
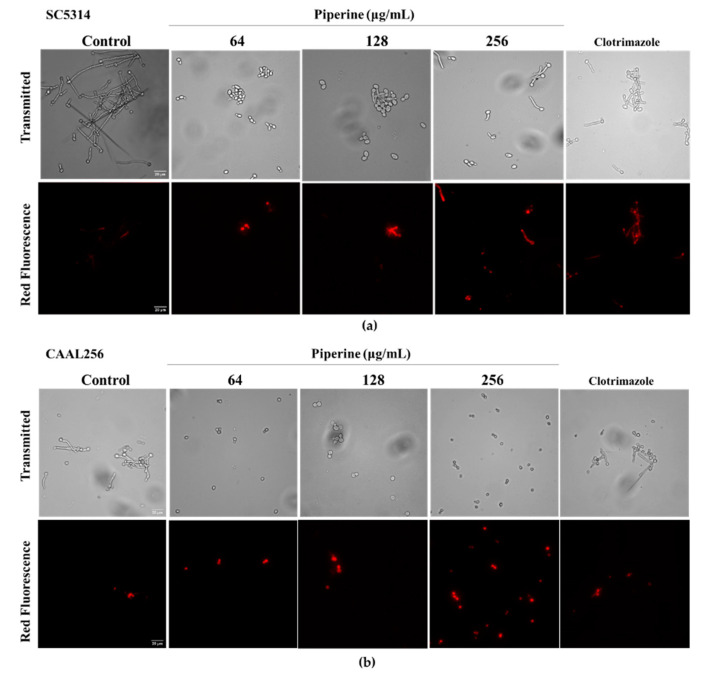
Piperine-induces apoptosis-like cell death of *C. albicans*. (**a**) SC5314 and (**b**) CAAL256 strains stained with Calcofluor-white (CW)/propidium iodide (PI). Control: untreated cells; and cells treated with 64, 128, and 256 µg/mL of piperine for 4 h at 37 °C. Positive control: clotrimazole 5 µg/mL. CW, excitation at 375 nm; PI, excitation at 555 nm. Scale bars, 20 µm.

**Figure 7 biomolecules-13-01729-f007:**
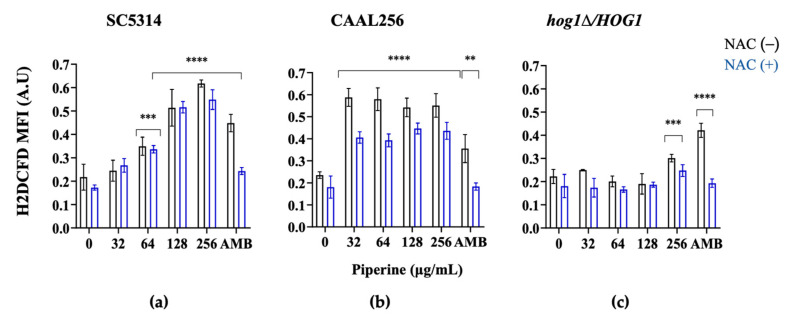
Piperine induces ROS accumulation on *C. albicans*. ROS production was assessed by cell staining with H_2_DCFDA; strains: (**a**) SC5314 (**b**) CAAL256 (**c**) *hog1∆*/*HOG1*. Cells were pretreated with NAC (+) (60 mM) or without NAC (–), followed by concentrations of piperine (32 to 256 μg/mL). MFI, mean fluorescence intensity; A.U, arbitrary units. AMB (amphotericin-B; SC5314: concentration 4 µg/mL and CAAL256: concentration 64 µg/mL) was used as positive control. The results are obtainable as means ± SD from three experiments. ** *p* ≤ 0.01, *** *p* ≤ 0.001, **** *p* ≤ 0.0001.

**Figure 8 biomolecules-13-01729-f008:**
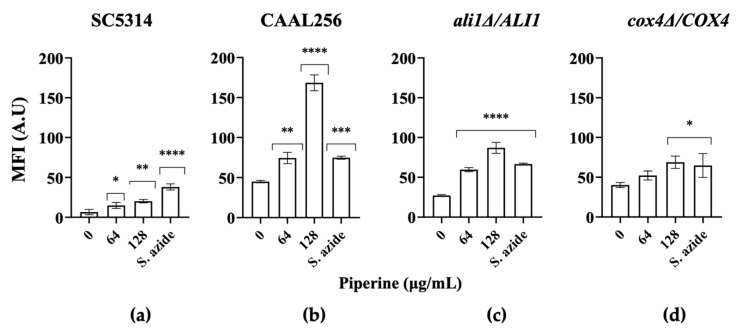
Piperine affects mitochondrial membrane potential on *C. albicans*. Membrane potential of mitochondria measured using rhodamine 123 fluorescence analysis in cells exposed to treatment with and without piperine; strains: (**a**) SC5314; (**b**) CAAL256; (**c**) *ali1Δ*/*ALI1* (**d**) *cox4Δ*/*COX4.* MFI, mean fluorescence intensity; A.U, arbitrary units. Sodium azide (NaN_3_) (5 mM) was used as positive control. Mitochondrial membrane depolarization showing mean ± SD from three experiments. * *p* ≤ 0.05, ** *p* ≤ 0.01, *** *p* ≤ 0.001, **** *p* ≤ 0.0001.

**Figure 9 biomolecules-13-01729-f009:**
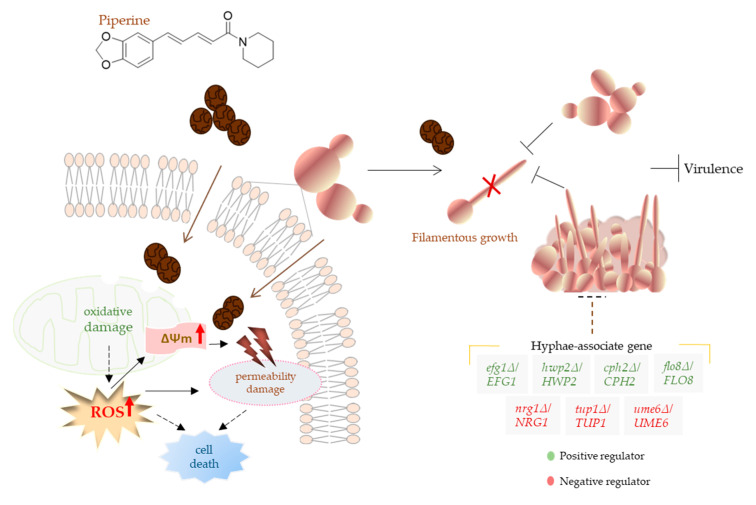
Schematic illustration: piperine’s antifungal mechanism against *C. albicans*. Piperine may displays its antifungal potential through two distinct pathways. It can impair mitochondrial function and generate ROS, ultimately leading to the breakdown of the cellular membrane permeability barrier and contributing to cell death (**left side**), or alternatively, it can target virulence factors (can exert antibiofilm and antihyphal activities) and influence essential pathways for *C. albicans* virulence (**right side**). Red cross indicate inhibition.

**Figure 10 biomolecules-13-01729-f010:**
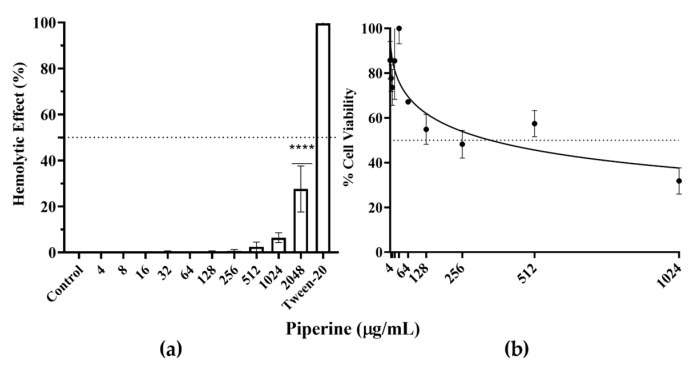
The in vitro cytotoxicity of piperine. (**a**) Hemolysis of human erythrocytes exposed using different concentrations of piperine for 1 h at 37 °C. (**b**) Cell viability of fibroblast (percentages of survival). Vehicle control: DMSO. Data were presented as the mean ± standard deviation of three replicate experiments. *p*-values of <0.05 were used to indicate statistical significance as follows: **** *p* ≤ 0.0001.

**Table 1 biomolecules-13-01729-t001:** Inhibitory effects of piperine against *Candida* spp.

*Candida* spp.		Piperine (μg/mL)	
	MIC_80_ (24 h)	MIC_80_ (48 h)	MFC (24 h)
*C. albicans* (SC5314) (ATCC) ^a^	1024	2048	>2048
*C. albicans* (CAAL256) ^b^	512	512	2048
*C. parapsilosis* (ATCC 22019) ^a^	1024	2048	2048
*C. tropicalis* (ATCC 1369) ^a^	512	1024	2048
*C. krusei* (ATCC 6258) ^b^	2048	>2048	>2048

^a^ Fluconazole-sensitive isolates. ^b^ Fluconazole-resistant isolates.

**Table 2 biomolecules-13-01729-t002:** Checkerboard test. Combination of piperine with FLC and NAC (µg/mL) against *C. albicans*.

***Candida* spp.**	**MIC of FLC**		**MIC of Piperine**			
	Alone	With piperine	Alone	With FLC	FICI	Outcome
SC5314	1	0.25	2048	256	0.375	synergy
CAAL256	32	16	1024	32	0.531	additive
	**MIC of NAC**		**MIC of Piperine**			
	Alone	With piperine	Alone	With FLC	FICI	Outcome
SC5314	25	25	2048	64	1.031	indifference
CAAL256	50	50	1024	64	1.031	indifference

## Data Availability

Data are contained within the article and Appendix A.

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
