# Peer review of "Exploring the Potential Mechanism of Action of Piperine against Candida albicans and Targeting Its Virulence Factors"

_biomolecules, 2023, doi:10.3390/biom13121729_

Round 1

Reviewer 1 Report

Comments and Suggestions for Authors

Dear Editor

I reviewed the manuscript entitled “Piperine from Piper nigrum: A Novel Approach to Combating Candida Resistance and Virulence”.  

Additional comments:

Dear Editor

I reviewed the manuscript entitled “Piperine from Piper nigrum: A Novel Approach to Combating Candida Resistance and Virulence”.   

Additional comments:

Abstract

Use the maximum number of words allowed by the journal.

I suggest concluding based on the results obtained.

Keywords

Use different words to those listed in the title.

Introduction

I consider that the introduction is well described. However, I suggest that first discuss the state of the art and finalize with a sentence of the main goal.

It is necessary to review the use of italics in scientific names of microorganisms. (In throughout the document)

Materials and methods

Include references in all methodologies used.

Results and discussion

Conclusion

The conclusion must be clear and concise, based on the results obtained in this study. Do not include speculations in this section on factors that were not analyzed in this study, instead, describe speculations in the discussion section.

It is advised to reduce similarity (<1 %), specifically with references with high-similarity content (references 1, 2, and 3). I suggest reducing this percentage as much as possible and resubmitting a corrected version. SEE THE ATTACHED FILE WITH THE OBTAINED REPORT FOR YOUR SUPPORT

Author Response

Reviewer 1

Comments and Suggestions for Authors

Thank you for the considerations and suggestions, in order to clarify the editor observations, we make the following comments and changes:

Abstract: Use the maximum number of words allowed by the journal

Answer: Thank you very much for the observation, the abstract was adjusted.

I suggest concluding based on the results obtained

Answer: We have concluded based on our results.

Keywords: Use different words to those listed in the title

Answer: Since the manuscript title has been altered, the keywords used remain suitable.

Introduction

I consider that the introduction is well described. However, I suggest that first discuss the state of the art and finalize with a sentence of the main goal.

Answer: Thank you for the observation, but we consider that the manuscript discusses first the state of the art: starting with Candida spp, followed by inhibition problems, and showing how plant extracts and metabolites are a good option for treatment. After showing how piperine, a natural compound, has demonstrated its antifungal effects, and we conclude with the objective of the study. However, the last sentence was modified to improve the observation made: Line 88 to 96.

It is necessary to review the use of italics in scientific names of microorganisms. (In throughout the document)

Answer: The changes were made (modifications in red)

Materials and methods: Include references in all methodologies used.

Answer: The references in the methodologies were included.

2.3 Purification of piperine by Centrifugal Partition Chromatography (CPC), was added: (Kanaki et al., 2018) and (Chen et al., 2009)

2.7 Yeast-to-Hyphal Transition Assay, was added: (Lee et al., 2018) and (Iadnut A. et al., 2019) and (Chaves et al., 2012)

Results and discussion

Conclusion

The conclusion must be clear and concise, based on the results obtained in this study. Do not include speculations in this section on factors that were not analyzed in this study, instead, describe speculations in the discussion section.

Answer: Thank you for the observation. the conclusion paragraph was modified. We do not include speculations or factors that were not analyzed in this section.

It is advised to reduce similarity (<1 %), specifically with references with high-similarity content (references 1, 2, and 3). I suggest reducing this percentage as much as possible and resubmitting a corrected version. SEE THE ATTACHED FILE WITH THE OBTAINED REPORT FOR YOUR SUPPORT

Answer: Thank you for the observation, the percentage of similarity was decreased.

Reviewer 2 Report

Comments and Suggestions for Authors

The study describes piperine, isolated from the fruits of Piper nigrum, which inhibited the proliferation of Candida species and even resistant strains. Although there are some studies on piperine and Candida in the literature, the authors sought to investigate the mechanisms of action of piperine and its effects on virulence factors in Candida albicans. Piperine suppressed the hyphal transition in both liquid and solid media, prevented biofilm formation, and resulted in cellular distortions that were observed under scanning electron microscopy (SEM) for both fluconazole-sensitive and fluconazole-resistant C. albicans strains. The results demonstrated that piperine treatment increased cell membrane permeability and disrupted mitochondrial membrane potential. Furthermore, it induced the accumulation of intracellular reactive oxygen species (ROS) in C. albicans. The manuscript explored some of the mechanisms related to the activity of piperine against C. albicans and the authors suggest the molecule as a new anti-virulence factor. The work is interesting, as in addition to being faced with the growing problem of resistance associated with Candida species. It was also very well contextualized and carried out using various techniques and methods to understand the action of piperine.

Author Response

We appreciate the comments, we did not find a special request, and we believe that with the adjustments made we did not change the paper in depth.

Reviewer 3 Report

Comments and Suggestions for Authors

The Author should consider also the following article recently publishe in Natura Product Research.

Antimicrobial, antileishmanial and cytotoxic compounds from Piper chaba

Tarannum Naz , Ashik Mosaddik , Md. Motiur Rahman , Ilias Muhammad , Md. Ekramul Haque & Somi Kim Cho

Natural Product Research, Volume 26, 2012 - Issue 11

Author Response

Reviewer 3

Comments and Suggestions for Authors

The Author should consider also the following article recently publishe in Natura Product Research.

Antimicrobial, antileishmanial and cytotoxic compounds from Piper chaba. Tarannum Naz , Ashik Mosaddik , Md. Motiur Rahman , Ilias Muhammad , Md. Ekramul Haque & Somi Kim Cho. Natural Product Research, Volume 26, 2012 - Issue 11

Answer: Thank you for your observation. Although this manuscript primarily centers on Piper chaba extract, the isolated compounds discussed in the article are 1 (Bornyl piperate) and 2 (piperlonguminine). However, compound 3, which is piperine, is mentioned only in the context of its isolation, without corresponding research results in bacteria or fungi, cytotoxicity or Antileishmanial activity of extracts, as reported in Tables 1, 2, and 3. Additionally, this is not a recent publication, but from 2012.

Reviewer 4 Report

Comments and Suggestions for Authors

The manuscript offers a new perspective regarding the potential use of piperine isolated from natural sources as antifungal agent. In my opinion, the quality of the manuscript recommends it as suitable for publication in present form. However, the following issues must be considered:

1.       Please, check again all Latin names and change them as italic (i.e. line 15 – Candida, line 18 – Piper nigrum)

2.       Line 106: Is it CdCl3?

Author Response

Reviewer 4

Comments and Suggestions for Authors

The manuscript offers a new perspective regarding the potential use of piperine isolated from natural sources as antifungal agent. In my opinion, the quality of the manuscript recommends it as suitable for publication in present form. However, the following issues must be considered:

  1. Please, check again all Latin names and change them as italic (i.e. line 15 – Candida, line 18 – Piper nigrum)

Answer: Thank you for your observation Latin names have been changed to italics and have been checked throughout the manuscript.

  1. Line 106: Is it CdCl3?

Answer: As reported in others manuscript: deuterated chloroform 99.8 atom % deuterium (CDCl3). (with the D in capital letters). Line 106 modified with the full name.

Round 2

Reviewer 3 Report

Comments and Suggestions for Authors

The authors' response to my request to include a reference is not satisfactory. It is an indication of a lack of attention to the literature.

I suggest rejecting this manuscript.